# Preschool Language Development of Children Born to Women with an Opioid Use Disorder

**DOI:** 10.3390/children8040268

**Published:** 2021-03-31

**Authors:** Hyun Min Kim, Reisha M. Bone, Brigid McNeill, Samantha J. Lee, Gail Gillon, Lianne J. Woodward

**Affiliations:** 1School of Health Sciences, University of Canterbury, Christchurch 8041, New Zealand; reisha.bone@canterbury.ac.nz (R.M.B.); samantha.lee@canterbury.ac.nz (S.J.L.); lianne.woodward@canterbury.ac.nz (L.J.W.); 2Child Well-Being Research Institute, University of Canterbury, Christchurch 8041, New Zealand; brigid.mcneill@canterbury.ac.nz (B.M.); gail.gillon@canterbury.ac.nz (G.G.); 3School of Teacher Education, University of Canterbury, Christchurch 8041, New Zealand

**Keywords:** opioid, methadone, CELF-P, language, child, outcome, neonatal abstinence syndrome

## Abstract

Increasing evidence suggests that prenatal exposure to opioids may affect brain development, but limited data exist on the effects of opioid-exposure on preschool language development. Our study aimed to characterize the nature and prevalence of language problems in children prenatally exposed to opioids, and the factors that support or hinder language acquisition. A sample of 100 children born to pregnant women in methadone maintenance treatment and 110 randomly identified non-exposed children were studied from birth to age 4.5 years. At 4.5 years, 89 opioid-exposed and 103 non-exposed children completed the preschool version of the Clinical Evaluation of Language Fundamentals (CELF-P) as part of a comprehensive neurodevelopmental assessment. Children prenatally exposed to opioids had poorer receptive and expressive language outcomes at age 4.5 years compared to non-opioid exposed children. After adjustment for child sex, maternal education, other pregnancy substance use, maternal pregnancy nutrition and prenatal depression, opioid exposure remained a significant independent predictor of children’s total CELF-P language score. Examination of a range of potential intervening factors showed that a composite measure of the quality of parenting and home environment at age 18 months and early childhood education participation at 4.5 years were important positive mediators.

## 1. Introduction

The use of prescribed and illicit opioids has increased dramatically in the last decade, particularly in the US, but also more recently in Australia, New Zealand and Europe [1,2,3,4,5]. As a consequence, opioid use during pregnancy has also increased which is concerning given the importance of the intrauterine environment for fetal and child development [6,7]. Thus, there has been growing interest in the effects of prenatal opioid exposure and other correlated pre and postnatal risk factors on both the short- and long-term outcomes for children born to women with an opioid use disorder (OUD). Existing evidence suggests increased risks for a range of adverse neonatal outcomes, including poorer fetal growth and Neonatal Abstinence Syndrome, as well as the possibility that child risks may extend into early and middle childhood [3,8,9,10,11,12,13]. However, longer-term follow-up studies are limited and remain somewhat inconclusive [10,12,14]. 

Language acquisition is an important developmental milestone during the early childhood years [15]. Language-specific speech perception typically begins to develop around 6 months postpartum, with vocabulary and syntactic abilities growing rapidly from age 1.5 to 3 years [16,17]. These skills are also critically important for children’s longer term social and educational development, with receptive and expressive oral language competencies being strongly predictive of children’s written language abilities and educational achievement at school, as well as their longer term academic and socioeconomic achievement during adolescence and early adulthood [18,19,20,21]. To date, few studies have examined the language development of children born to women with OUD, with findings often mixed due to methodological issues such as small sample sizes, high attrition, and the measurement of language abilities using global developmental measures such as the Bayley Scales of Infant and Toddler Development or older measures such as the Reynell Developmental Language Scales [22,23,24,25,26,27,28,29]. Nonetheless, existing evidence does tend to suggest that opioid-exposed children may be at increased risk of language delay [23,24,25,26,30,31,32]. Thus, there is a need for well-controlled investigations examining the nature of these children’s language difficulties using psychometrically sound, standardized measures of early language performance such as the Clinical Evaluation of Language Fundamentals—Preschool (CELF-P) [33]. Such information is important to help guide early education and health practices to better support the needs of these children and their families.

Another important issue concerns the mechanisms that may place opioid-exposed children at an increased risk of language difficulties, and even more importantly, the identification of both (a) prenatal factors that may help explain later child risks, and (b) modifiable postnatal factors which might buffer them from later risk. On average, research shows that pregnant women who use/abuse opioids are more likely to come from lower socioeconomic backgrounds [24,25], to have fewer educational qualifications, and to be subject to higher rates of other mental and physical health problems [24,25,34,35,36]. Greater physical and mental stress among these women can translate to poorer prenatal care, which coupled with prenatal opioid exposure and associated poorer prenatal nutrition, may negatively impact fetal development [37,38]. These factors may also have an indirect effect on child language development through family functioning and parenting [24,25,39,40].

Postnatally, in addition to these ongoing influences, for some of these women and infants, additional family challenges correlated with maternal opioid use/abuse may further impede their ability to provide the sensitive and rich linguistic interactions and experiences that are most conducive to fostering a child’s early language development [26,30,32]. Incarceration [40] and involvement with child protection services is also more common [26,40,41], with these problems often being intergenerational and limiting women’s access to family social supports [26,40]. For example, findings suggest that children raised in families affected by parental drug use are exposed to lower levels of responsive parenting and less stimulating learning environments [26,35,40,42], factors that are relevant for language development.

Although the impact of these risk factors on children’s development is quite well-documented, less is known about the potential buffering role of positive postnatal family and preschool experiences on mitigating these environmental risks. There is good evidence that parents and families can also greatly support their children’s language development by providing culturally enriched and stimulating environments that facilitate rich language interactions inside and outside the home [26,30,35,43,44,45]. Engagement in early childhood education has also been shown to improve child language outcomes [46].

Collectively, these findings raise the need to consider the effects of other adverse prenatal exposures, family socioeconomic circumstances and children’s postnatal family life and environmental experiences on the language [23,24,25,26,27,30,31] and other developmental outcomes of opioid-exposed (OE) children [36,40,47]. Thus, the aims of this study were as follows. First, to characterize the language outcomes at age 4.5 years of children prenatally exposed to opioids relative to their non-opioid exposed (NE) same-age peers. Language will be assessed using the CELF-P [33] which assesses a range of receptive and expressive language abilities including linguistic concepts, sentence structure, sentence recall, word structure and label formulation. Second, to examine the extent to which observed between-group differences in language development can be explained by the effects of confounding factors correlated with maternal opioid use/abuse during pregnancy. Third, to identify parenting, family functioning and early childhood education/intervention factors that might mitigate the risk of language problems in children prenatally exposed to opioids. We hypothesized that OE children would have poorer language outcomes compared to NE children at 4.5 years and that this difference would remain after controlling for the effects of other prenatal confounding risk factors. Finally, we hypothesized that protective family and environmental factors such as positive parent–child interactions and early childhood education (ECE) center attendance would reduce the adverse effects of prenatal opioid exposure on language development at 4.5 years. These aims and the corresponding hypothesized pathways are illustrated in Figure 1.

## 2. Materials and Methods

### 2.1. Study Participants

The participants were drawn from a prospective longitudinal study of two groups of children born between 2003 and 2008 at Christchurch Women’s Hospital, New Zealand [11,13,34]. The first study group consisted of 100 infants (58 male) born to women with OUD who were maintained on methadone during pregnancy. The second reference or control group of children recruited were not exposed to opioids, and were randomly selected from the hospital booking schedule during the same period. The reference or non-exposed (NE) group consisted of 110 infants (48 male) who were born into families whose socioeconomic profiles were representative of the regional population at the time of birth [11,14]. Mothers from both groups were recruited during the third trimester or at birth. Exclusion criteria included inability or refusal to consent, HIV diagnosis, delivery outside the region, very preterm birth (gestation of 32 weeks or less), suspected fetal alcohol syndrome, congenital anomalies, and non-English speaking. Retention to age 4.5 years was 89% for the OE group and 94% for the NE group. Figure 2 provides the overall study participation and retention from birth to 4.5 years. Further details about recruitment and study women pregnancy course and personal histories are available in previous publications [11,34]. Key family social background information and infant neonatal characteristics are summarized in Table 1.

### 2.2. Procedure

Infant perinatal characteristics and medical treatment history were extracted from hospital records. Detailed information about mothers’ family social backgrounds, pregnancy nutrition, physical and mental health was also collected as part of a comprehensive maternal interview in the late third trimester or at birth. As shown in Figure 2, each study child and their primary caregiver were invited to participate in a detailed child neurodevelopmental evaluation and family assessment at ages 18 months, 2 years and 4.5 years. The 18-month follow-up consisted of a home visit to allow additional contextual information on parenting and the child’s home environment to be collected. Our primary language outcome measure was administered by research staff blinded to children’s group assignment at age 4.5 years as part of a half-day neurodevelopmental assessment. All study protocols were approved by the Upper South B Regional Ethics Committee, Canterbury, New Zealand (Ref: URB/07/10/042). Written informed consent was obtained from all primary caregivers at each assessment point. Key study measures used in this analysis are briefly described below.

### 2.3. Study Measures

#### 2.3.1. Child Language Outcomes at Age 4.5 Years

Children’s language development was assessed at age 4.5 years using the CELF-P UK [33]. The CELF-P is suitable for children aged 3 to 6 years. It consists of six subtests that were combined to form standardized composite scores for receptive language, expressive language and total language ability. The receptive language index measures the listening comprehension ability and is computed based on the sum of scores on three subtests: (1) linguistic concepts, a measure of the child’s ability to interpret spoken directions and concepts related to logical operations; (2) sentence structure, the ability to interpret spoken sentences; and (3) basic concepts, the knowledge of concepts such as number, size, direction and location. The expressive language index is an overall measure of expressive language skills and is computed based on the sum of scores of three subtests: (1) recalling sentences in context, a measure of the child’s ability to imitate and repeat spoken sentences; (2) formulating labels, the ability to name objects, actions, etc.; and (3) word structure, the ability to apply word structure rules and use appropriate pronouns. The scores on each subtest were scaled using the test age-equivalent population-norms and converted to standardized scores with a mean of 10 and standard deviation (SD) of 3. Finally, the CELF-P total language score was created by combining children’s composite receptive and expressive language scores as described in the manual. Composite expressive and receptive scores and total language scores were standardized to have a mean of 100 and SD of 15.

The CELF-P UK was standardized on 588 British children aged 3 to 7 years, who were representative of the UK population in terms of gender, race/ethnicity, socioeconomic status, and geographical region. The UK version was used as the Australian and New Zealand version had not been developed and was not available at the time of the assessment. The version used has good construct validity, with moderate intercorrelations (0.37 to 0.56) between subtests suggesting that they each measure a distinct, yet related, language skill. It is also internally consistent, with composite score reliability coefficients ranging from 0.76 to 0.91 across age groups. The subtest intercorrelations and internal reliability coefficients were also in line with those from the larger US normative sample [33]. Overall language delay was defined as a total CELF-P score greater than one SD below the reference group mean. The cut-points derived from the reference group data aligned well with the test norms, and the results obtained from the analysis of total and each subscale scores were similar regardless of which criteria were applied. Table 2 describes children’s performance in CELF-P.

#### 2.3.2. Potential Prenatal Confounding Factors

##### Prenatal Risk Exposures

The extent of prenatal stress and risk exposures was measured using a composite measure of several factors. These included (a) the extent to which pregnant mothers used (and the infant was exposed to) other licit and illicit substances; (b) extent of maternal depressive symptoms during pregnancy reported by mothers at birth; and (c) the quality of maternal pregnancy nutrition. A cumulative risk score indexing the overall exposure to the above three factors was computed by first transforming all factors into a common scale, reverse-coded when necessary, then taking the average of those values. A brief description of each of the measures contributing to this overall prenatal risk index is provided below.

(a)Other maternal licit and illicit substance use during pregnancy was assessed using three independent measures to determine the extent of each child’s poly-substance exposure. First, detailed information about mothers’ substance use was collected as part of a comprehensive maternal interview completed in the late third trimester or at birth. All mothers reported the frequency and duration of tobacco, alcohol, cannabis, opiates, benzodiazepine, and stimulant use for each pregnancy trimester. Second, women in the methadone group provided random urine samples over the course of their pregnancies that were analyzed for the presence of illicit substances (cannabis, opiate, benzodiazepine and stimulants). Finally, meconium samples were collected from a subsample of methadone-exposed (81%) and comparison (46%) infants at birth. The results from the maternal urine and infant meconium tests were used to assess the reliability of maternal self-reported data on their use of cannabis, benzodiazepines, stimulants, opiates and antidepressants. This analysis showed strong concordance (>80%) between toxicological and self-report data. Based on this combined information, for the purposes of this analysis, a diversity score was computed to describe the total number of different types of licit/illicit substances other than opioids, that women used during their pregnancy.(b)The extent of maternal depressive symptoms during pregnancy was measured either in the late third trimester or at birth using the Edinburgh Postnatal Depression Scale (EPDS) [48]. The EPDS is a widely used screening tool for depressive symptoms and contains 10 statements that describe symptoms such as “I have felt sad or miserable” and “I have been so unhappy that I have been crying”. The women were asked to rate each item using a 4-point Likert-type scale based on their symptomology during the last two weeks. Each item can be scored 0 to 3 with 3 indicating a greater extent of depressive symptoms. Scale internal consistency in this sample was reasonable at alpha = 0.79.(c)A self-reported measure of the nutritional quality of each study woman’s diet during pregnancy was collected as part of the third trimester/birth maternal interview. Women were questioned about their weekly intake of fruit, vegetables, meat, milk, bread, other cereals, and eggs. An overall quality of maternal pregnancy nutrition score was estimated from the average total number of weekly servings consumed from each of these food groups.

##### Child Sex

Children’s biological sex was recorded at birth and included in this analysis given known sex differences in early language and pre-literacy skill development [49].

##### Level of Maternal Education

Maternal educational attainment was correlated with maternal opioid use and other parental social background variables such as single parenthood, being a young parent and having a lower socioeconomic status. After analyzing each of their bivariable associations with language outcomes, maternal education was selected to represent the social background as the variable had the highest effect size when used to predict language development. Maternal educational attainment was measured on a 6-point scale ranging from 1 to 6, with 1 being “left school before age 16 years without any qualifications” and 6 being “attainment of university degree”.

#### 2.3.3. Protective Postnatal Family and Environmental Factors

In order to capture the extent of responsive parenting, cultural/social enrichment and stimulation in a child’s environment at 18–24 months, and to test our hypothesis that positive parenting and environmental factors might mitigate language delay risk, a composite variable was created using items from parent interview and direct observational data. A wide range of postnatal measures were considered. These spanned (a) parental mental health and ongoing drug use, (b) parenting, (c) family functioning, (d) the quality of the home environment, (e) child physical health including vision and hearing problems, and (f) children’s early childhood education attendance and any early intervention exposures. The final set of measures were selected based on their mediating effects and representation of non-overlapping yet related aspect of parenting and family environment.

First, the overall level of exposure to stimulating environment and activities was measured using a 22-item, 5-point Likert-type scale that measures children’s exposure to different types of activities. For example, “be read a story” with responses ranging from 0: “never” to 4: “daily”. The original scale was designed for New Zealand families by the Dunedin Multidisciplinary Health and Development Study [50]. See Appendix A for the individual items.

The infant and toddler Home Observation for Measurement of the Environment (HOME) [51] was also used in the composite to provide a measure of caregiver acceptance of the child. Of the six HOME subscales, the acceptance subscale, a measure of nonpunitive or non-intrusive parenting practice, was chosen as it contained items that were non-overlapping with the customized cultural capital measure. Higher scores on this scale indicated greater parental acceptance for the child’s behavior with fewer instances of verbal and physical punishment. The internal reliability measured using Cronbach’s alpha was 0.74 for the sample.

Finally, a direct observational measure of mother–child emotional connection during a free and structured play session at age 18–24 months was included. This dyadic rating was completed by trained assessors blinded to child history and group membership, with inter-rater reliability assessed between independent raters for at least 20% of all observations and found acceptable with intraclass correlation coefficient estimates in the range of 0.784–0.961.

The overall composite measure of positive parenting and environmental factors at 18-months–2 years was constructed using the above three measures by transforming the scores into a common scale then taking the average of the scores. A composite rather than individual scores was used in the analysis to assess the cumulative effect, improve estimation efficiency and obtain accurate estimates of variability.

#### 2.3.4. Early Childhood Education Center Attendance at Age 4.5 Years

Participation in ECE at age 4.5 years was measured based on the maternal report of average number of hours a week the child attended the ECE center.

Table 3 provides a summary of variables included in the analysis along with a selection of variables that were considered as potential intervening factors.

### 2.4. Statistical Analysis

Maternal and infant characteristics were first summarized using mean and SDs for continuous variables and the percentage of each sample for categorical variables using either independent samples t-test or chi-squared test of independence (results shown in Table 1). Second, between-group differences in children’s CELF-P scores were then compared using a similar approach (see Table 2).

Third, we examined the extent to which between-group differences might be explained by confounding factors. The set of potential confounding variables was selected based on (a) previous research findings, (b) temporal considerations (i.e., during pregnancy) relative to the opioid-exposure and the outcome, and (c) the strength of associations between each of these measures and opioid-exposure and our primary language outcomes. Potential confounding variables were then analyzed by examining their associations with the exposure and then again with the language outcome. Key covariates included the prenatal risk composite, comprised of prenatal exposure to other substances, maternal depression and the quality of nutrition during pregnancy, child sex and maternal educational attainment.

Fourth, once covariates were refined and included in the analysis, we then examined the intervening or mediating role of a similarly refined set of postnatal factors. This was done by first examining whether opioid-exposure predicted the candidate mediator, and then again whether the mediator predicted the study outcome [52]. The mediation analysis was performed using multiple regression [53]. The model selection was aided by the explained sum of squares measure adjusted for multiple regressors and the mean squared error (MSE) statistic computed using the 10-fold cross validation technique. Descriptive statistics and regression analyses were performed using R version 4.0.3 (The R Foundation, Vienna, AT) and SPSS version 26 (IBM Corp., NY, USA), and alpha = 0.05 defined statistical significance.

## 3. Results

### 3.1. Child Language Outcomes at Age 4.5 Years

Table 2 describes children’s performance on the standardized CELF-P test at age 4.5 years. Across all subscales and composites, OE children scored lower on average than NE children. OE children scored more poorly in receptive (*p* < 0.001), expressive (*p* < 0.001) and overall language scores (*p* < 0.001) indicating pervasive delays/deficits in language competencies compared to their NE peers. This was further reflected in rates of overall language delay, with OE children being almost three times more likely than NE children to meet criteria for significant language delay (29% vs. 11%).

### 3.2. Effects of Potential Confounding Factors

The results in Table 2 suggest that prenatal opioid exposure is associated with pervasive language difficulties, and an increased risk of delayed language development before school entry. However, it is also possible that these observed language difficulties may, either in part or in full, reflect the effects of other confounding factors correlated with maternal opioid use during pregnancy rather than the direct effects of opioid exposure.

To examine this issue, the between-group differences in OE and NE children’s overall language scores were adjusted for the effects of child sex, maternal education and exposure to other prenatal risk factors. The results from this analysis are shown in the covariate adjusted model in Table 4. They show that although the inclusion of the prenatal risk composite in multivariate models significantly reduced the association between prenatal opioid exposure and children’s total CELF-P scores at age 4.5 years (*p* = 0.04), it did not fully explain the between-group difference in OE and NE children’s total language scores (*p* = 0.03).

### 3.3. Role of Family and Environmental Factors

From a wider range of potential mediating and moderating factors, the key quality of environment and parent-child relationship factors that mediated the effects of prenatal opioid exposure on children’s language development at age 4.5 years were selected based on the criteria described in the Methods. Between-group comparisons in Table 3 show that OE children were being raised in postnatal family environments characterized by lower levels of nurturant parenting, and had fewer opportunities to experience stimulating activities both within and outside the home as toddlers. Their overall score on the combined protective environment variable was significantly lower (*p* < 0.001) than that of their NE peers. The average weekly hours of attendance at the ECE centers were also lower for the OE children (*p* = 0.01) compared to the NE children at age 4.5 years.

To examine whether more positive parenting, family functioning and early education indicated by these factors can potentially buffer children from developing preschool language problems, a mediation analysis was performed. Using multiple regression, a mediation model was constructed using the group indicator and both sets of confounding and mediating variables described above as the explanatory variables. Then, the model was used to examine the effects of mediating factors on the between-group differences in the total language scores. The results of the mediation model are described in Table 4 and Figure 3.

As shown, results confirm the study hypothesis that together these protective factors fully mediated the relationship between prenatal opioid exposure and preschool language development. None of the intervening factors were moderators based on analyses of interaction terms. An assessment of model fit also demonstrated the superior fit of the mediation model (adjusted R^2^ = 0.36, MSE = 203.73) compared to either the bivariable (adjusted R^2^ = 0.24, MSE = 241.23) or the covariate adjusted (adjusted R^2^ = 0.29, MSE = 227.18) models, indicating that the mediation model better explains the exposure and language outcome relationship given the number of predictors. The direct effect of opioid exposure on language was 6.27, while the indirect effect was 11.08, resulting in the total effect of 17.35 and mediation of 20.3% of the differences. Finally, both the protective parenting and environmental factor at 2 years (*p* < 0.001) and weekly hours of attendance at the ECE center at age 4.5 years (*p* = 0.01) were significant mediators.

## 4. Discussion

Study findings demonstrate that women treated with methadone for an OUD are characterized by multiple risk factors that at least in part contribute to the poorer preschool language outcomes of their children. However, our findings also show that positive postnatal factors such as sensitive parenting, enriched cultural environment and ECE can to some extent reduce the adverse effects of prenatal opioid exposure on preschool language development. The results that children born to women with OUD have, on average, poorer receptive, expressive and overall language abilities than children non-exposed to opioids are consistent with limited but increasing evidence that OE children may be at higher risk of language delays compared to their NE peers [9,23,24,25,26,30,31,32]. However, where previous research has mostly focused on the global assessment of development, and language considered as an auxiliary measure, this study utilized a well-established and appropriately normed measure of preschool language development.

A consideration of relevant sociodemographic variables on preschool language development also revealed the following. Although female children obtained better language scores in a bivariable association model, this sex difference was no longer statistically significant once other risk factors were considered in the multivariate model. Previous research has indicated that male children may be more susceptible to the adverse effects of prenatal substance exposure and consequently achieve poorer developmental outcomes [47,54,55]. The results in this study suggest that child sex may have developmental implications, but neural and environmental mechanisms through which it influences language development are complex as illustrated by the mixed findings in a recent review [49], and more research is needed in order to understand these mechanisms in the context of prenatal opioid exposure.

The level of maternal education is an important indicator of the quality of cognitive stimulation and early learning environment a child experiences in their daily life. Not surprisingly, a higher level of maternal education was positively associated with children’s language outcomes at 4.5 years. However, this effect was attenuated once other risk factors were taken into account, which indicates that the educational background alone does not negate the impact of opioid exposure on children’s language development. The multiple risk factors that accompany opioid exposure such as maternal depression, poor nutrition and exposure to other substances continue to impede language development even after adjusting for positive environmental factors at age 2 and 4.5 years. Thus multi-domain wrap-around support may be needed to improve the overall quality of pre- and antenatal conditions for mothers with a substance use disorder. Although previous research has looked at the importance of multiple risk factors on the development of children prenatally exposed to opioids, few studies to date have investigated the additive effects of these risks on language outcomes.

More importantly, the role of sensitive parenting, a stimulating home environment and ECE attendance has not been investigated previously with a well-controlled, reasonably sized, and regionally representative sample. The results of the mediation analysis indeed reveal the striking role of parenting, daily stimulation and ECE on OE children’s language development. These environmental factors when combined, fully mediated the clear disadvantages in language development of OE children compared to their NE peers. The importance of caregiver warmth and positive dyadic relationships on language development has also been shown in an earlier study [26].

The positive effect of non-intrusive and nurturant parenting and a culturally enriched environment on children’s language development seen here is especially important considering the developmental trajectory of these high-risk children. A recent study using the same population of children studied here has shown persistent developmental risks across multiple domains of psychosocial development [14]. These findings combined with those from the present study highlight the need for early identification and support for these children and their families to optimize these children’s developmental opportunities. Language and preliteracy skills strongly predict later literacy and educational success [18,19,21]. Therefore, providing appropriate support for caregivers and access to high quality ECE may further help in buffering these children from long-term educational and associated social disadvantages.

Previous research has also suggested that the different developmental patterns between OE and NE children become more apparent around age 18–24 months when language abilities become increasingly more important in many of the cognitive developmental tasks [30]. Both this observation and our results suggest that targeted support in early childhood would be important in mitigating the risks of language delay and potential literacy problems. A very limited number of the studies that have looked at the outcomes of home-based interventions for families affected by drug dependence showed some improvement in child neurodevelopment and suggested cautious optimism around the efficacy of these measures [36,56,57]. The results of our study provide a clear rationale for further studies aimed at identifying the most effective and appropriate ways to support sensitive parenting and improving ECE outcomes for these high-risk children and families.

Despite some compelling results, this study is not without limitations. Given our necessarily observational research design, study results reflect associations rather than a direct cause and effect. Although the sample size is larger than most previous studies, the choice of confounders and mediators were carefully weighed in order to achieve the most parsimonious model appropriate for testing the hypothesized pathways. Hence, some important confounding variables such as the role of fathers and family stability may not have been included in the model. Despite our best efforts to follow-up on all cases and the rather impressive retention rate for a very high-risk population, there were missing responses, which may have introduced bias in the estimation. However, a careful examination of key participant characteristics for previous research suggests that the observations are likely to be missing at random [14].

Overall, this study adds valuable insight into preschool language development of OE children in a well characterized and largely unselected cohort of children born to women with OUD, with excellent sample retention. Findings suggest that these children are at higher risk of language delay at school entry relative to their same-age typically developing peers. The mechanism leading to later language problems is complex, representing the cumulative effects of a range of adverse prenatal exposures correlated with maternal OUD, family social background factors, and children’s postnatal rearing environments. Importantly, the results show that sensitive, non-intrusive parenting and engagement in ECE can help to buffer these children against early language difficulties. Further follow-up of the longer-term language-related outcomes of this high-risk group of children is needed, along with studies examining the effects of early intervention strategies aimed at better supporting these children and families both antenatally and postnatally.

## Figures and Tables

**Figure 1 children-08-00268-f001:**
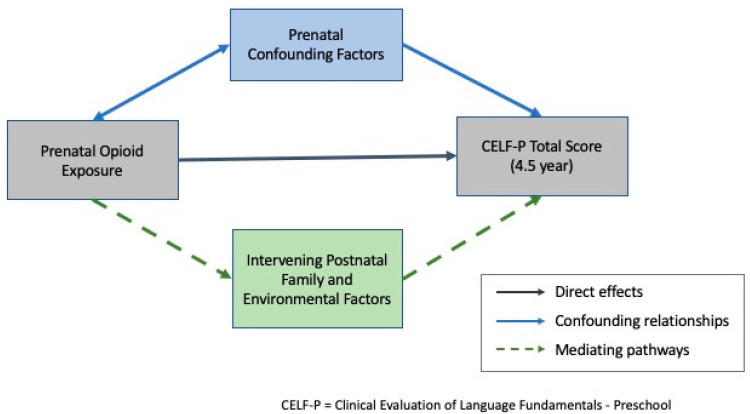
Hypothesized pathways.

**Figure 2 children-08-00268-f002:**
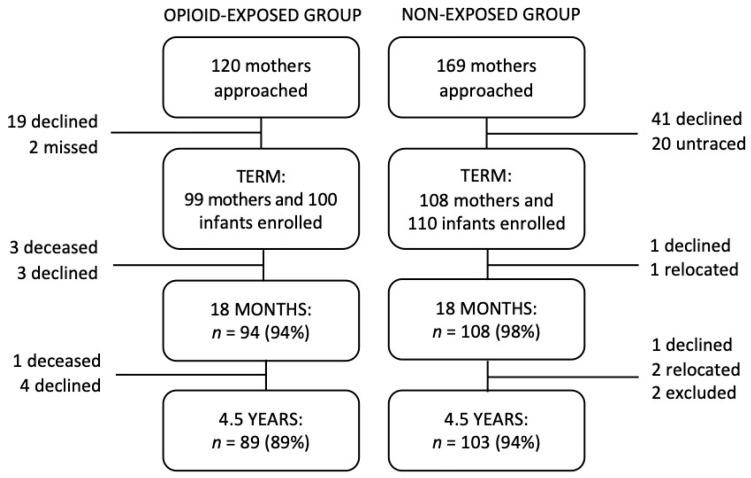
Participant recruitment and retention from birth to 4.5 years. *n* = number of participants.

**Figure 3 children-08-00268-f003:**
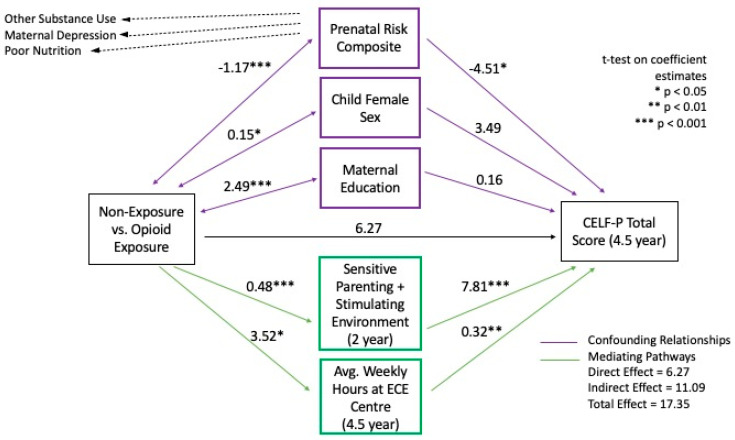
Mediation model.

**Table 1 children-08-00268-t001:** Maternal and infant characteristics.

Measure	Opioid- Exposed (*n* = 89)	Non-Exposed (*n* = 103)	*p* ^1^
Maternal characteristics at birth			
Young mother, *n* (%)	3 (3.4)	5 (4.9)	0.620
Minority ethnicity, *n* (%)	21 (23.6)	18 (17.5)	0.290
Maternal schooling level, mean (SD)	1.44 (1.11)	3.85 (1.94)	<0.001
Low socioeconomic status, *n* (%)	72 (81.8)	23 (22.3)	<0.001
Single parent, *n* (%)	43 (48.9)	10 (9.7)	<0.001
Maternal wellbeing during pregnancy			
Average methadone dose mg/d, mean (SD)	62.25 (35.08)	0.0 (0.0)	<0.001
Nutrition during pregnancy score, mean (SD)	56.49 (20.75)	90.82 (25.52)	<0.001
Maternal depression symptoms, mean (SD)	13.87 (6.09)	6.91 (4.81)	<0.001
Any cigarette use, *n* (%)	82 (92.1)	16 (15.7)	<0.001
Any alcohol use, *n* (%)	15 (16.9)	21 (20.6)	0.510
Any cannabis use, *n* (%)	42 (47.2)	1 (1.0)	<0.001
Any benzodiazepine use, *n* (%)	24 (27.0)	0.0 (0.0)	<0.001
Any stimulant use, *n* (%)	18 (20.2)	0.0 (0.0)	<0.001
Any additional opioid use, *n* (%)	23 (25.8)	0.0 (0.0)	<0.001
Prenatal risk composite, mean (SD)	0.62 (0.59)	−0.58 (0.44)	<0.001
Infant neonatal characteristics			
Female, *n* (%)	36 (40.9)	58 (56.3)	0.030
Gestational age in weeks, mean (SD)	38.78 (1.78)	39.23 (1.71)	0.090
Preterm (<37 weeks gestation), *n* (%)	10 (11.4)	7 (6.8)	0.270
Birthweight in grams, mean (SD)	3058.79 (470.15)	3412.85 (586.74)	<0.001
Body length (cm), mean (SD)	50.17 (3.08)	51.73 (5.25)	0.020
Pharmacologic treatment for neonatal abstinence syndrome, *n* (%)	78 (87.6)	0 (0.0)	<0.001

^1^*t*-test or Chi-squared test of independence. *n* = number of participants; SD = standard deviation.

**Table 2 children-08-00268-t002:** Prenatal opioid exposure and language development at age 4.5 years: CELF-P scores.

CELF-P Scores at 4.5 Years	Opioid- Exposed (*n* = 89)	Non- Exposed (*n* = 103)	t	*p*
Receptive language composite, mean (SD)	83.57 (14.56)	100.00 (17.51)	−6.93	<0.001
Linguistic concepts, mean (SD)	6.20 (2.92)	9.65 (3.22)	−7.66	<0.001
Basic concepts, mean (SD)	8.17 (3.05)	10.36 (2.30)	−4.94	<0.001
Sentence structure, mean (SD)	7.53 (3.06)	10.27 (3.41)	−5.75	<0.001
Expressive language composite, mean (SD)	85.24 (15.08)	100.77 (14.31)	−7.24	<0.001
Recalling sentences, mean (SD)	6.93 (3.09)	9.89 (3.36)	−6.25	<0.001
Formulating labels, mean (SD)	7.36 (2.61)	10.02 (2.70)	−6.85	<0.001
Word structure, mean (SD)	8.07 (3.45)	10.66 (2.90)	−5.61	<0.001
Total language score, mean (SD)	83.62 (14.85)	100.46 (15.97)	−7.45	<0.001
Delayed total language, *n* (%)	29 (32.6)	11 (10.7)	13.89 ^1^	<0.001 ^1^

^1^ Chi-squared test of independence. CELF-P = Clinical Evaluation of Language Fundamentals-Preschool; *n* = number of participants; t = t-statistics.

**Table 3 children-08-00268-t003:** Intervening child, parenting and environmental factors from age 18 months to 4.5 years.

Intervening Factors.	Opioid- Exposed (*n* = 89)	Non- Exposed (*n* = 103)	t	*p*
Child factors (18mth)				
Any ear infection, *n* (%)	31 (35.6)	48 (46.6)	2.34 ^1^	0.126 ^1^
Any hearing problems, *n* (%)	1 (1.1)	3 (2.9)	0.71 ^1^	0.399 ^1^
Any vision problems, *n* (%)	10 (11.5)	16 (15.5)	0.65 ^1^	0.420 ^1^
Sleep disturbances, *n* (%)	14 (16.1)	18 (17.6)	0.08 ^1^	0.776 ^1^
Maternal illicit drug use (18 mth), *n* (%)	42 (47.2)	5 (4.9)	45.82^1^	<0.001 ^1^
Protective factors composite (2 yr)	−0.22 (0.68)	0.26 (0.54)	−5.18	<0.001
Acceptance—HOME (2 yr)	6.62 (1.28)	6.99 (1.05)	−2.19	0.030
Nurturance—Parent-Child (2 yr)	15.09 (2.84)	16.38 (2.60)	−3.22	0.002
Cultural capital (2 yr)	63.62 (7.06)	67.88 (5.78)	−4.56	0.001
ECE hours a week (4.5 yr)	18.53 (9.26)	21.89 (9.23)	−2.50	0.010

^1^ Chi-squared test of independence. *n* = number of participants; mth = month; yr = year.

**Table 4 children-08-00268-t004:** Mediation analysis of prenatal opioid exposure and language development at age 4.5.

Model	Predictor	Coef. Est. (95% CI)	*p*
Bivariable			
	Group: Non-Exposed	17.35 (12.74, 21.96)	<0.001
Adjusted R^2^ = 0.241, MSE = 241.23			
Covariate Adjusted			
	Group: Non-Exposed	7.87 (0.78, 14.97)	0.03
	Prenatal Risk Composite	−4.74 (−9.22, −0.25)	0.04
	Child Sex: Female	4.18 (−0.33, 8.70)	0.07
	Maternal Education	1.32 (−0.13, 2.77)	0.07
Adjusted R^2^ = 0.287, MSE = 227.18			
Mediation Model			
	Group: Non-Exposed	6.27 (−0.50, 13.03)	0.07
	Prenatal Risk Composite	−4.51 (−8.76, −0.25)	0.04
	Child Sex: Female	3.49 (−0.82, 7.78)	0.11
	Maternal Education	0.16 (−1.30, 1.62)	0.83
	Protective Factor Composite (2 yr)	7.81 (4.06, 11.57)	<0.001
	Hours a Week of ECE Attendance (4.5 yr)	0.32 (0.08, 0.56)	0.01
Adjusted R^2^ = 0.359, MSE = 203.73			
% Mediated = 20.33			

Coef. Est. = coefficient estimate; MSE = mean squared error.

## Data Availability

The data presented in this study are available on request from the corresponding author. Some restrictions may apply. The data are not publicly available due to privacy and ethical concerns.

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
