# Peer review of "Preschool Language Development of Children Born to Women with an Opioid Use Disorder"

_children, 2021, doi:10.3390/children8040268_

Round 1
Reviewer 1 Report
The authors present a thoughtfully derived study that explores language development in opioid exposed children, and concludes that this population is more likely to have delayed language at 4.5 years. The inclusion of other factors traditionally not included, such as home observations, protective factors, maternal depression and education etc. and the inclusion of a mediation model is noteworthy, as are the relatively large retention rates in both groups. There are a few major concerns:
- How do the authors conclude that opioid exposure, and not the other exposures (nicotine, THC, benzo, stimulants) are not the exposures more likely to inform language concerns? It might be informative in a conclusion to note an association rather than cause and effect, or perhaps that opioid exposure is a marker of risk factor for impaired language development in children. If not, the potential misuse of data such as these is the association of methadone, a medication commonly used for the treatment of OUD during pregnancy, with language delay; information which could deter women from necessary treatment at critical perinatal time periods.
- Was there any measure of maternal substance use during the study period? i.e. were all mothers stable in methadone maintenance or is there any information to inform continued substance use during critical periods of language development? Continued maternal use of substances during early childhood could portend a completely different effect on child development from exposures, with language more likely than other developmental domains to be affected.
- The language used to describe the OE group is somewhat pejorative: please consider women with opioid use disorders (person first language) rather than opioid-dependent. Referring to population as these women or these children can compound this perception, please adjust language.
- Minor concerns:
- Were other substances (i.e. cocaine, PCP, binge alcohol) considered as exposures? If no why not?
- Were other maternal psychopathologies other than depression explored? Information about maternal depression does not appear to be reported in the MS.
- Why was very preterm birth (<32 weeks) chosen as opposed to preterm birth? How was preterm birth defined in Table 1?
- How was non-exposed status in the NE group confirmed? Only maternal interview? It appears that exposures for the OE group might have been documented with urine toxicology testing (?); was this true for the NE group? If no, there could be a misrepresentation of the true rates of other substance use between groups as there is different data collection by group. Meconium for both groups was collected, at least partially, what were the results of this analysis?
- Is treatment for NAS meant to mean pharmacologic treatment?
All in all a nice effort that could inform the need for early intervention for language development in opioid exposed children.
Reviewer 2 Report
This is a thoughtful study that both identifies an important association between prenatal opioid exposures and later language skills and also identifies significant mediating factors. This will be an important addition to the literature. However, I do have a few minor edits that could improve the readability of the study.
Introduction:
This section is too long. Consider condensing and also eliminate run on sentences.
Intro 3rd paragraph is one sentence. Consider condensing with the paragraph following.
Final paragraph reads like it was lifted directly from a grant specific aims page. Need to condense.
Methods
Any incentives for recruitment? Or incentives for retention? Retention rate of 90% is super impressive. How was that attained?
What does “any additional opioid use” mean? Is this referring to opioids like heroin?
For table 1, would be helpful to have the scoring metrics in a footnote for nutrition score, prenatal risk composite; also unclear how depression symptoms are scored when looking at table alone.
Sentence stating “The UK version was used as the Australian and New Zealand version was not available at the time of the assessment” is unclear. Was UK version used because NZ version not developed yet?
Table 2 scoring metrics would also be helpful
Also unclear what the meconium samples were used for. Is that to confirm the presence of illicit substances?
Results
Figure 3 unclear what the *, ** and *** are signifying. I think it is p-values but I am unsure
Discussion
Need to highlight the finding about the complete mediation of effects by parenting in the first paragraph of the discussion. I think this is the most interesting finding as it implies we can change a child’s developmental trajectory to negate the influence of prenatal opioids.
Limitations paragraph talks about building a parsimonious model. Please elaborate on other factors that were left out but may be of significance.
Second to last paragraph is one sentence. Shorten and merge with last paragraph
